# Ultrathin Struts Drug-Eluting Stents: A State-of-the-Art Review

**DOI:** 10.3390/jpm12091378

**Published:** 2022-08-25

**Authors:** Attilio Leone, Fiorenzo Simonetti, Marisa Avvedimento, Domenico Angellotti, Maddalena Immobile Molaro, Anna Franzone, Giovanni Esposito, Raffaele Piccolo

**Affiliations:** Department of Advanced Biomedical Sciences, University of Naples Federico II, 80131 Naples, Italy

**Keywords:** coronary artery disease, drug-eluting stents, percutaneous coronary intervention, ultrathin struts

## Abstract

New-generation drug-eluting stents (DESs) represent the standard of care for patients undergoing percutaneous coronary intervention (PCI). Recent iterations in DES technology have led to the development of newer stent platforms with a further reduction in strut thickness. This new DES class, known as ultrathin struts DESs, has struts thinner than 70 µm. The evidence base for these devices consists of observational data, large-scale meta-analyses, and randomized trials with long-term follow-up, which have been conducted to investigate the difference between ultrathin struts DESs and conventional new-generation DESs in a variety of clinical settings and lesion subsets. Ultrathin struts DESs may further improve the efficacy and safety profile of PCI by reducing the risk of target-lesion and target-vessel failures in comparison to new-generation DESs. In this article, we reviewed device characteristics and clinical data of the Orsiro (Biotronik, Bülach, Switzerland), Coroflex ISAR (B. Braun Melsungen, Germany), BioMime (Meril Life Sciences Pvt. Ltd., Gujarat, India), MiStent (MiCell Technologies, USA), and Supraflex (Sahajanand Medical Technologies, Surat, India) sirolimus-eluting stents.

## 1. Introduction

Percutaneous coronary intervention (PCI) is the technique most frequently used to treat flow-limiting coronary artery stenoses. PCI is a continually evolving field, primarily as a result of the development, refinement, and iterations of technologies and devices [1]. The transition from early-generation to new-generation stents entailed a wide range of design changes in the metallic stent platform and its geometry [2,3]. While bare metal stents (BMSs) are usually made of stainless steel, most new-generation drug-eluting stents (DESs) consist of a cobalt-chromium alloy (CoCr) or a platinum-chromium alloy (PtCr) metallic platform. These improvements enabled the reduction of the strut thickness from 130–140 μm to 60–80 μm, resulting in greater deliverability and faster endothelial coverage following implantation. Differences among the platforms exist also with respect to the antiproliferative drug and the drug load. Newer stents use a lower drug load to facilitate the endothelisation process with a “limus” derivate as an antiproliferative drug (i.e., sirolimus, everolimus, zotarolimus, novolimus, biolimus, umirolimus). DESs were initially developed to reduce neointimal hyperplasia and restenosis ensuing after PCI with BMSs. However, when compared with BMSs, the use of early-generation DESs was associated with an increased risk of stent thrombosis (ST) [4]. To address this issue, new-generation DESs were designed with improved biocompatibility of the permanent polymer coating or with biodegradable polymer along with better safety of the released drug [5]. New-generation DESs outperformed BMSs in numerous endpoints, including cardiac death, myocardial infarction (MI), ST, and repeat revascularization [6,7]. As a result, new-generation DESs are currently recommended for all patients undergoing PCI irrespective of the anticipated duration of dual antiplatelet therapy [8]. Over the past few years, refinements in device technology prompted a further reduction of strut thickness with the introduction of a new class called “ultrathin strut DESs”. Despite the lack of a standardized definition, ultrathin DESs are defined as stents with a strut thickness less than 70 µm. The potential benefits of these device include deliverability, reduced vessel injury, and side branches’ flow disturbance. In this article, we provided a comprehensive overview of ultrathin DESs that are currently available ***(***Figure 1***)***.

## 2. Orsiro

The Orsiro coronary stent (BIOTRONIK, Bülach, Switzerland) consists of an ultrathin strut, cobalt-chromium platform, with a bioresorbable, sirolimus-eluting polymer (biodegradable polymer sirolimus-eluting stent (BP-SES)). It is available in diameters ranging from 2.25 to 4.0 mm and in lengths between 9 and 40 mm. Stents with diameters of 2.25 to 3.0 mm have a strut thickness of 60 μm, whereas stents with diameters of 3.5 to 4.0 mm have a strut thickness of 80 μm [9]. The Orsiro BP-SES consists of different layers. The innermost layer is a cobalt-chromium alloy (the PRO-Kinetic energy™ stent) arranged in a double-helix pattern, designed to improve deliverability by lowering the crossing profile. Due to the proBIO coating, the metallic platform of the stent is not in direct contact with the blood vessel or bloodstream. This is an amorphous-hydrogen-rich silicon carbide coating bonded to the metallic platform. The proBIO coating is a specific feature of this device, and it may have positive effects related to a lowered rate of metallic stent corrosion and less tissue inflammation, including allergic reactions to the metal. Of interest, this “passive shield” offered by the proBIO coating is permanent (i.e., the coating does not delaminate over time). The outer layer is made of a bioabsorbable poly-l-lactic acid (PLLA) polymer containing sirolimus. The active BIOlute™ coating is distributed asymmetrically with a thickness of 7.5 μm on the abluminal side and a thinner, 3.5 μm layer on the luminal portion of the stent. The sirolimus load is 1.4 μg/mm^2^. The PLLA degrades over 2 years, releasing 50% of the drug within 30 days and 80% during the first three months. The safety and efficacy of the Orsiro SES has been evaluated in a number of clinical trials involving multiple clinical settings (acute and chronic coronary syndromes) and subsets of lesions (de novo, small vessel disease, chronic total occlusions, in-stent restenosis).

The BIOFLOW I was the first-in-man trial evaluating the Orsiro BP-SES in patients with single de novo coronary artery lesions and showed excellent results in terms of 9-month late lumen loss (LLL) [10]. The BIOFLOW II trial showed the non-inferiority of the Orsiro BP-SES compared with a durable polymer (DP) everolimus-eluting stent (Xience EES Abbott Vascular, Santa Clara, California) in terms of target-lesion failure (TLF) and lower mortality in vessels from 2.25 to 2.75 mm, suggesting a potential benefit of ultrathin struts up to 5-years follow-up [11,12]. The BIOFLOW IV trial was designed for regulatory submission in Japan and confirmed the non-inferiority of the Orsiro BP-SES with the Xience EES among 575 patients with de novo lesions [13]. The BIOFLOW V was designed to test the performance of Orsiro in all-comers PCI patients across 13 countries. A total of 1334 patients were randomly assigned to either Orsiro or Xience in a 2:1 ratio. About 50% presented with an acute coronary syndrome. At 1-year follow-up, the Orsiro SES outperformed the Xience EES, demonstrating consistently lower clinical event rates in TLF (*p* = 0.0399) and significantly lower rates of target-vessel myocardial infarction (TV-MI) (*p* = 0.0155). There results were confirmed up to 5-years follow-up. In order to improve statistical significance, the authors of BIOFLOW V combined the results with those of the BIOFLOW II and IV trials with a Bayesian approach, reporting a posterior probability for non-inferiority of 100% and a posterior probability of superiority of 97% for the Orsiro [14,15]. The BIOSCIENCE was a non-inferiority trial that randomized 2119 patients (3139 lesions) to receive either the Orsiro SES or the Xience EES. The Orsiro was shown to be not inferior in terms of the primary endpoint of TLF at 12 months (p-non-inferiority < 0.0004). As a novel finding, a subgroup analysis of patients treated for ST-segment elevation myocardial infarction (STEMI) showed a lower risk of TLF with the Orsiro SES than the Xience EES at 12 months (*p* = 0.024) [16]. However, this superiority was not confirmed by the BIOSCENCE 5-year follow-up, where no differences were found between the two stents in terms of TLF, suggesting that the advantage of ultrathin struts and biodegradable polymer may decrease after complete degradation of the polymer and endothelial healing [17]. An individual, patient-level, meta-analysis of five randomized trials (BIOFLOW-II, BIOFLOW-IV, BIOFLOW-V, BIOSCIENCE, and BIOSTEMI) showed a similar risk of TLF among 5780 patients randomly allocated to BP-SES or DP-EES up to 5-years follow-up [18]. The BIO-RESORT trial randomized patients to receive one of three stents: the Orsiro BP-SES, the Synergy BP-EES (74 μm), or the Resolute Integrity DP zotarolimus-eluting stent (91 μm). The trial showed the non-inferiority of the Orsiro BP-SES at 1-, 2-, and 3-years follow-up [19,20,21]. In the BIONYX trial, the Orsiro BP-SES served as the control stent for the newly designed Resolute Onyx DP-ZES. No differences were found among the two devices in clinical outcomes, although definite or probable stent thrombosis was less frequent in the experimental arm [22]. Conversely, the Orsiro BP-SES was associated with a significantly lower risk of definite or probable stent thrombosis and TLF against the Nobori and the Biofreedom biolimus-eluting stents, respectively, in the SORTOUT VII and IX trials. This difference in clinical outcomes could be explained by the thicker struts (120 μm) of the two platforms made of stainless steel [23,24].

## 3. Coroflex ISAR Neo

Coroflex ISAR (B. Braun, Melsungen, Germany) is a polymer-free, cobalt-chromium, sirolimus-eluting stent. The stent platform is based on the CX-Blue Ultra stent for 2.0 to 3.0 mm diameters (55 µm) and on the CX-Blue Neo stent for 3.5 to 4.0 mm diameters (65 µm). The polymer-free matrix is contained on the abluminal aspect of the microporous stent surface and consists of sirolimus at a concentration of 1.2 µg/mm^2^ and probucol to control the drug release. Probucol serves as matrix-builder and is a highly lipophilic, lipid-lowering agent, with antioxidant effects [25]. Approximately 80% of sirolimus is released within 30 days, while the process is completed at 90 days. The device is available in diameters ranging from 2.0 to 4.0 mm and lengths between 9 and 38 mm. The Coroflex has been tested in the Intracoronary Stenting and Angiographic Results: Test Efficacy of Sirolimus- and Probucol- and Zotarolimus Eluting Stents (ISAR-TEST-5) study, a non-inferiority trial including 3002 patients. According to the primary endpoint, a composite of cardiac death, target-vessel-related myocardial infarction, or target lesion revascularization (TLR), the Coroflex ISAR stent was non-inferior to the Resolute ZES (P-non-inferiority = 0.006; P-superiority = 0.74) [26]. These findings were confirmed at 5- and 10-years follow-up with a low incidence of probable/definite ST in both groups and consistent results through pre-specified subgroups of age, gender, diabetes mellitus, and vessel size [27,28].

## 4. Biomime, Biomime Morph, and Evermine 50

BioMime (Meril Life Sciences Pvt. Ltd., Gujarat, India) is an ultrathin (65 µm), cobalt-chromium, biodegradable polymer, sirolimus-eluting stent. The device presents a hybrid design with closed cells at both ends and open cells in the middle, potentially favouring a better stent expansion and lesser likelihood of edge dissection. The open cell design in the mid-part of the stent should also facilitate side branch access and treatment. The BioPoly biodegradable polymer has a low thickness (~2 μm), is composed by poly-l-lactic acid (PLLA) and poly-d,l-lactide-co-glycolide (PLGA), and degrades in approximately 60 days. The sirolimus concentration is 1.25 µg/mm^2^ and is released over 30–40 days after stent implantation. The BioMime SES is available in lengths from 8 to 48 mm and diameters from 2.00 to 4.50 mm.

The MeriT-1, MeriT-2, and MeriT-3 trials established the safety and efficacy of the Biomime SES in treating single de novo and complex coronary lesions [29,30]. In the meriT-1 study, a first-in-human, single-centre trial, the Biomime SES showed a low LLL at 8-months angiographic follow-up in 30 patients (30 lesions) [29]. The meriT-2 trial was a larger, single-arm study including 250 patients (355 lesions) with a higher prevalence of diabetes and multivessel disease. At 1-year follow-up, major adverse cardiac events (MACE) occurred in 8.9% of patients [31]. The meriT-3 study included 1161, all-comers patients undergoing PCI with Biomime SES across 15 centres in India and showed a low rate (2.35%) of MACE at 1-year follow-up [30]. The meriT-V trial was the first to randomly compare in a 2:1 ratio the Biomime SES with the Xience EES among 256 patients. At 9-months angiographic follow-up, the Biomime SES resulted in being non-inferior to the Xience EES with respect to the primary endpoint of LLL [32].

The Biomime Morph (Meril Life Sciences Pvt. Ltd., Gujarat, India) is a further iteration of the Biomime SES technology featuring a tapered stent system with two different proximal and distal diameters (e.g., 2.75–2.25 mm; 3.00–2.50 mm, etc). The tapered stent system together with the long available lengths (30, 40, 50, 60 mm) allows the treatment of diffuse, long lesions.

The Evermine 50 (Meril Life Sciences Pvt. Ltd., Gujarat, India) presents the same hybrid cell stent design, but the cobalt-chromium platform is thinner (50 µm). The stent releases everolimus, which is loaded with a concentration of 1.25 µg/mm^2^.

## 5. Mi Stent

The MiStent sirolimus-eluting stent (MiStent SES) (MiCell Technologies, Durham, NC, USA) is an ultrathin (64 μm), biodegradable polymer, sirolimus-eluting stent. The sirolimus is built in the vessel wall as microcrystals, and its crystalline form enables a controlled drug release. Indeed, the polymer (PLGA) is reabsorbed within 3 months after implantation, minimizing the risk of vascular inflammation, while the sirolimus is continuously delivered up to 270 days.

The Mistent has been evaluated in the Dessolve trials. Dessolve I was the first-in-man study and enrolled 30 patients with de novo lesions. At 18-months follow-up, the primary endpoint of LLL was attested to be 0.08 mm. Of interest, 27 underwent optical coherence tomography, which showed complete strut coverage [33]. In the Dessolve II trial, 184 patients were randomized 2:1 to Mistent vs. Endeavor ZES. No difference was found between groups in terms of TLF up to 5-years follow-up [34]. In the Dessolve III trial, the MiStent SES was non-inferior to the Xience EES with respect to the primary endpoint of cardiac death, target-vessel MI, or clinically indicated target lesion revascularization at 12 months among 1398 all-comers patients enrolled across 20 European centres [35].

## 6. The Supraflex Family

The Supraflex (Sahajanand Medical Technologies, Surat, India) system is an ultrathin strut (60 μm) stent made of a cobalt-chromium alloy and a biodegradable polymer releasing sirolimus. The stent diameters range from 2.0 to 4.5 mm and the lengths from 8 to 48 mm. Sirolimus has a concentration of 1.4 μg/mm^2^, and the polymer gradually degrades over 9–12 months. Approximately 70% of the drug is released within 7 days. The latest iteration of the Supraflex is the Supraflex Cruz with two long dual-Z connectors from “valley to valley” between the struts, to enhance deliverability and increase the flexibility of the stent and a re-designed proximal shaft to allow a better pushability. The Supraflex was first evaluated in a large-scale, multicentre observational registry. The FLEX registry included 995 patients (1242 lesions) in nine Indian centres and reported a low rate (3.7%) of MACE at 1-year follow-up (3.7%) [36]. The S-FLEX UK registry was conducted across different U.K. centres and showed a low rate of TLF (2.4%) and no definite stent thrombosis among 469 patients undergoing PCI with the Supraflex SES [37]. The TALENT trial randomly compared the Supraflex SES with the Xience EES among 1430 patients. At 12-months follow-up, there was no difference between the groups for the primary endpoint, a composite of cardiac death, target-vessel myocardial infarction, or clinically indicated TLR [38]. Although the trial was not powered for all-cause mortality, a significantly higher mortality rate was found in the experimental arm (2.0% vs. 0.6%), which might be due to the play of chance. At 2- and 3-years follow-up, the primary endpoint occurred at similar rates in the Supraflex SES and Xience EES arms (6.9% vs. 7.9%, *p* = 0.491; 8.1% vs. 9.4%, *p* = 0.406, respectively) [39,40]. Several ongoing studies will provide evidence on the performance of Supraflex Cruz SES in different settings, including acute coronary syndromes, multivessel-disease and high-bleeding-risk patients [41,42]. The FIRE trial (Clinical trial.gov: NCT03772743) is an all-comers, prospective, randomized, multicentre trial, using the SUPRAFLEX/SUPRAFLEX Cruz SES to evaluate the outcomes of a functionally driven complete revascularization in elderly patients with MI and multivessel disease [34].

The trial has completed the enrolment phase. Differently, the ongoing multivessel TALENT trial will compare clinical outcomes between the SUPRAFLEX Cruz SES and the SYNERGY EES, in 1550 patients with three-vessel disease. The primary endpoint is a composite of all-cause death, stroke, myocardial infarction, or any repeat revascularization, whereas the secondary endpoints include a superiority comparison of the SUPRAFLEX Cruz SES versus the control arm at 24 months [35].

Eventually, the Cruz HBR registry will enrol 1200 patients to prove that the Supraflex Cruz is not inferior to the BioFreedom stent in HBR patients with respect to a device-oriented composite endpoint (DOCE) at 1 year.

## 7. Benefit of Strut Thickness Reduction: A Class Effect?

Ultrathin struts have been implemented to further enhance PCI outcomes. In recent years, a number of large-scale randomized trials have assessed their potential benefits. In addition, two large-scale systematic meta-analyses have been performed to investigate the differences between the available ultrathin platforms. Bangalore et al. conducted a meta-analysis of >11,500 patients. Out of 10 trials included, 8 evaluated the Orsiro SES (5444 patients), 1 the Mistent SES (703 patients), and 1 the Biomime SES (170 patients) [43]. At 1-year follow-up, the ultrathin strut DES reduced by 16% the risk of TLF compared with a conventional new-generation DES (relative risk, 0.84; 95% CI, 0.72–0.99). The results were consistent across trials, and no differences emerged according to the type of ultrathin DES used. The risk reduction in TLF was driven by lower rates of MI, mainly attributed to a lower rate of ST and periprocedural MI. These findings were confirmed by a subsequent larger meta-analysis including 16 randomized trials with 20,701 patients [44]. The ultrathin DESs were the Orsiro (12 trials, 17,658 patients), the MiStent (2 trials, 1582 patients), the BioMime (1 trial, 256 patients), and the Supraflex (1 trial, 1435 patients). At a mean follow-up of 2.5 years, ultrathin-strut DESs were associated with a lower risk of TLF and TVF. There was no significant interaction according to stent type in the ultrathin strut group.

Because the stent comparators in these meta-analyses were all new-generation DESs with biocompatible polymers, the observed differences might be due to the reduction of greater than 10 μm in strut thickness [45,46]. This difference may potentially enhance strut endothelization due to a reduction in vessel injury and vascular inflammation and reduced periprocedural MI due to less flow disturbance to the side branches [47].

## 8. Ultrathin Stents in High-Risk Subgroups

### 8.1. STEMI

STEMI setting is burdened by an increased risk of early ST due to the prothrombotic milieu of the culprit lesions. At 12- and 24-months follow-up, the BIOSCIENCE trial demonstrated lower rates of TLF in STEMI patients receiving the Orsiro SES than in patients receiving the Xience EES (3.3% vs. 8.7%, *p* = 0.024 at 12 months; 5.4% vs. 10.8%, *p* = 0.043 at 24 months) [48,49,50]. The BIOSTEMI trial was specifically designed to demonstrate the superiority of the Orsiro SES in patients with STEMI. At 12 months, the Orsiro SES resulted in lower rates of TLF compared with the Xience EES (4% vs. 6%, posterior probability of superiority = 0.986) [17]. The experience with the Supraflex Cruz stent from multicentre word registries demonstrated initial favourable outcomes in different subsets of lesions and clinical and patient characteristics [36]. The Supraflex SES proved safe and effective in 229 STEMI patients from the Talent trial and 198 from the Flex registry with a low incidence of TLR and stent thrombosis [36,38].

### 8.2. Chronic Total Occlusions

The performance of Orsiro in CTO lesions was evaluated in the Prison IV trial [51]. A total of 330 patients were randomized in a 1:1 ratio to the Orsiro SES or the Xience EES. Although underpowered for clinical outcomes, the trial failed to show the non-inferiority of the Orsiro SES in terms of LLL at 9 months. These results were confirmed at 3-years follow-up with a higher rate of MACE in the Orsiro SES arm. Anyway, the subgroup analysis of patients with CTO included in the BIOFLOW III and SORT-OUT VII trials showed a low rate of TLF and TLR [52]. Out of 185 patients with CTO treated with Supraflex Cruz SES from the Flex registry, the rate of MACE at 1 year was as low as 6.6% [36].

### 8.3. Diabetes Mellitus

In an analysis of the Talent trial, the rate of DOCE in diabetic patients was 5.8% in the Supraflex arm vs. 8.5% in the Xience arm. The 1-year clinical outcomes with the Orsiro DES in diabetic patients were assessed in a patient-level pooled analysis of the diabetic population from the BIOFLOW II, IV, and V trials. A similar rate of 1-year TLF was observed among 494 patients treated with the ultrathin BP-SES and 263 patients treated with the thin-strut DP-EES (6.3% vs. 8.7%) [53]. Similar results were found in a subgroup analysis of the SORT OUT VII trial [54]. In the ISAR-TEST 5 Trial, out of 3002 patients, 28.7% treated with the Coroflex ISAR Neo were diabetics; across such a subgroup, the outcomes were consistent with those of non-diabetic patients up to 10-years follow-up [28].

### 8.4. Small Vessel Disease

Small vessel coronary artery disease is common among patients undergoing PCI, and myocardial revascularization in this subset remains challenging owing to an increased risk of restenosis and technical failure. Evidence in this field is limited for new-generation DESs, including ultrathin struts DESs in view of a lack of dedicated trials. As such, the evidence is mainly limited to subgroup or post hoc analyses of randomized trials [55]. In the BIORESORT trial, out of 3514 patients, 1506 were in the small vessel subgroup (defined as vessels ≤2.75 mm). Patients treated with Orsiro experienced a lower rate of TLR in comparison with the other treatment groups [56]. In contrast, in the Bioscience trial, where small vessel disease was defined as a vessel diameter ≤3 mm, no difference was observed in the 5-year rate of TLF between the BP-SES and DP-EES groups [57]. No outcome difference was observed in a prespecified analysis according to the vessel size in patients treated with the Coroflex ISAR Neo in the ISAR-TEST 5 [28]. Patients from the TALENT trial had small vessel disease in 44.9% of cases with a rate of DOCE of 8% at 1 year, not significantly different from patients treated with Xience.

### 8.5. In-Stent Restenosis

While data about new-generation DESs for the treatment of in-stent restenosis showed improved performance in comparison to drug-eluting balloons [58], data on the performance of ultrathin DESs in this setting are lacking. When compared with the drug-coated balloon in the BIOLUX trial, Orsiro resulted in being non-inferior in terms of LLL and TLF. However, further data are necessary in this subgroup of patients [59].

### 8.6. Limitations of Ultrathin DES

When a high radial force is required, such as CTO or calcific lesions, the presence of the ultrathin struts might potentially reduce the performance of the stent in terms of stent expansion. However, specific data on this issue are scant. Of interest, stent expansion capacity is more limited with ultrathin DESs than other new-generation DESs, and therefore, their use in large vessels may be challenging [60].

## 9. Future Directions

The search for the ideal stent continues, enhancing research in improving DESs design and their performance in real-word challenging settings. The outcomes of the previous generation DESs, along with the impact of specific PCI techniques, ancillary techniques (e.g., intracoronary imaging), and structural features, have been investigated in different complex clinical scenarios such as left main stem disease, in-stent restenosis, and coronary bifurcation lesions [61,62,63]. Differently, data regarding the safety and efficacy of ultrathin struts DESs in left main or coronary bifurcations are still scarce. Once these safety data in real-world cohorts become available, it is likely that ultrathin stents will be the standard of care for most revascularization procedures. We may anticipate that, in the near future, research in stent design will focus on the development of novel and more biocompatible drugs, alloys, and polymers. Future DESs are also supposed to be characterized by a progressive improvement in deliverability and flexibility. Whether such enhancements will entail further strut thickness reduction largely depends on the evidence that modification of DES structure would not come at the cost of an insufficient radial force, as discussed above.

## 10. Conclusions

The introduction of ultrathin struts DESs constituted a further iteration in the field of PCI technology with the potential to further hone the safety and efficacy profile of PCI. The results of multiple studies enrolling a huge number of patients and providing long-term follow-up may make ultrathin struts stents the preferred stent strategy in several clinical scenarios and lesion subsets. Ongoing randomized trials will increase evidence on the efficacy and safety of ultrathin DES.

## Figures and Tables

**Figure 1 jpm-12-01378-f001:**
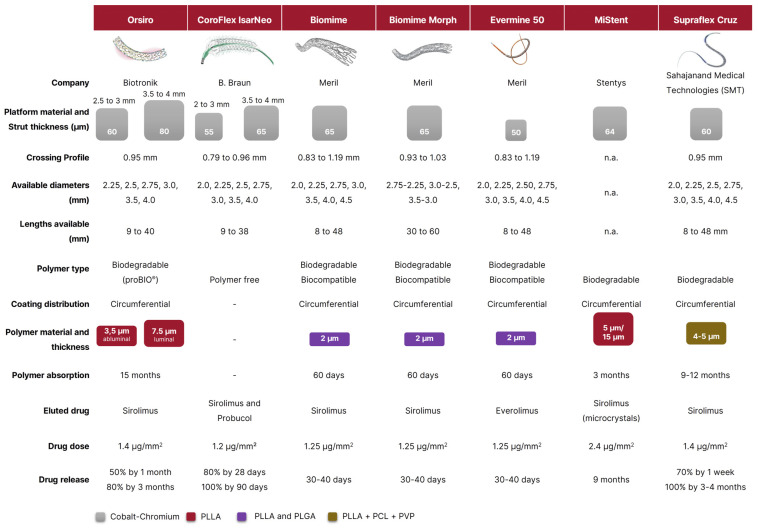
Comparison of currently available ultrathin drug-eluting stents (DESs). CoCr: cobalt chrome; n.a.: not available, PCL: poly-caprolactone; PLGA: poly-d,l-lactide-co-glycolide; PLLA: poly-l-lactic acid; proBIO: amorphous hydrogen-rich silicon carbide; PVP: poly vinyl pyrrolidone.

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
