# Peer review of "Ultrathin Struts Drug-Eluting Stents: A State-of-the-Art Review"

_jpm, 2022, doi:10.3390/jpm12091378_

Round 1

Reviewer 1 Report

The authors reported a state of art review about the current evidence and devices features of the available ultrathin struts Drug-Eluting Stents. Device characteristics and clinical available data were reported for the Orsiro, the CoroFlex Isar Neo, the Biomime, the Biomime Morph, the Evermine, the MiStent and the Supraflex platforms. The topic is of interest and the review is comprehensive. The paper is nicely written and the manuscript has an interesting structure, with headings referring to the evidence for each investigated devices and sub-headings referring to specific clinical scenarios that may be of interest to a large number of readers.

The reviewer has the following comments:

-        As per editorial request, and to further improve readers’ interest I suggest adding a paragraph to explore “Future directions”. In particular, I suggest that Authors should acknowledge the paucity of data regarding the performance os such devices in real-word scenarios and complex settings as left main stem disease, in-stent restenosis and coronary bifurcation lesions. The outcomes of the previous generation DES (namely very-thin strut DES, that is DES with strut thickness < 100 um) was investigated in some of these contexts along with the impact of specific PCI techniques and of ancillary techniques such as intracoronary imaging, with interesting results (i.e. see DOI:  10.1016/j.amjcard.2019.02.013 and DOI: 10.1161/CIRCINTERVENTIONS.119.008325).

-        I deem the paragraph “Benefit of strut thickness reduction, a class effect?” nice and of interest. However, as this is a review focusing on DES features, Authors should more appropriately investigate current evidence (if any) about structural features differentiating the investigated devices such as numbers of crowns and connectors, along with their potential impact on outcomes. The impact of such features was indeed investigated for the previous generation of DES, with fascinating results (see DOI: 10.1002/ccd.28667).

-        The modern PCI era is characterized by an interesting debate about the offsetting relevance of PCI ancillary techniques, namely invasive coronary physiology vs intracoronary imaging, mainly in optimizing procedural results and therefore patients’ outcomes. This particularly concern peculiar setting as ACS ( please see : European Heart Journal, Volume 42, Issue 45, 1 December 2021, Pages 4656–4668;  Catheter Cardiovasc Interv. 2020 Jun 1;95(7):1259-1266. doi: 10.1002/ccd.28410; Int J Cardiol. 2017 Oct 1;244:54-58. doi: 10.1016/j.ijcard.2017.05.108). As experts in this field Authors should take a position on whether the DES design will have a role in such debate, as reduction of strut thickness has been associated with the concern of inadequate radial force that may likely influence stent expansion and apposition.

-        In the section “Limitation of Ultrathin DES” the authors state that “stent expansion capacity is more limited with ultrathin DES than other new-generation DES”. Please provide some reference or hypothesis to support such statment.

Minor comments:

There are minor spelling mistakes, please consider careful proof reading before re-submission ) i.e. page 15 “ such has CTO or calcific lesions” should be “as”; in the abstract “ has struts thinner than less than” should be has struts thinner than.

·        Please control list of abbreviation. In the section “Coroflex IsarNeo” the authors used for the first time the term TLR without providing the extended version. 

Author Response

Reviewer #1

The authors reported a state of art review about the current evidence and devices features of the

available ultrathin struts Drug-Eluting Stents. Device characteristics and clinical available data were

reported for the Orsiro, the CoroFlex Isar Neo, the Biomime, the Biomime Morph, the Evermine, the

MiStent and the Supraflex platforms. The topic is of interest and the review is comprehensive. The

paper is nicely written and the manuscript has an interesting structure, with headings referring to the

evidence for each investigated devices and sub-headings referring to specific clinical scenarios that

may be of interest to a large number of readers.

We thank the Reviewer for the careful and thoughtful review of our manuscript.

  1. As per editorial request, and to further improve readers’ interest I suggest adding a paragraph to

explore “Future directions”. In particular, I suggest that Authors should acknowledge the paucity of

data regarding the performance of such devices in real-word scenarios and complex settings as left

main stem disease, in-stent restenosis and coronary bifurcation lesions. The outcomes of the

previous generation DES (namely very-thin strut DES, that is DES with strut thickness < 100 um) was

investigated in some of these contexts along with the impact of specific PCI techniques and of

ancillary techniques such as intracoronary imaging, with interesting results (i.e. see

DOI: 10.1016/j.amjcard.2019.02.013 and DOI: 10.1161/CIRCINTERVENTIONS.119.008325).  

We thank for this comment and akcnowledge the criticism raised by the reviewer.  In this revised version of the manuscript, we added a new section named “Future directions”.

  1. I deem the paragraph “Benefit of strut thickness reduction, a class effect?” nice and of interest.

However, as this is a review focusing on DES features, Authors should more appropriately investigate

current evidence (if any) about structural features differentiating the investigated devices such as

numbers of crowns and connectors, along with their potential impact on outcomes. The impact of

such features was indeed investigated for the previous generation of DES, with fascinating results

(see DOI: 10.1002/ccd.28667).

We thank the Reviewer for this comment. We carefully described structural features of the investigated device in the dedicated section. However, in the updated version of the manuscript, we tried to address this issue in the “ Future directions” paragraph.

  1. The modern PCI era is characterized by an interesting debate about the offsetting relevance of PCI

ancillary techniques, namely invasive coronary physiology vs intracoronary imaging, mainly in

optimizing procedural results and therefore patients’ outcomes. This particularly concern peculiar

setting as ACS ( please see : European Heart Journal, Volume 42, Issue 45, 1 December 2021,

Pages 4656–4668; Catheter Cardiovasc Interv. 2020 Jun 1;95(7):1259-1266. doi:

10.1002/ccd.28410; Int J Cardiol. 2017 Oct 1;244:54-58. doi: 10.1016/j.ijcard.2017.05.108). As

experts in this field Authors should take a position on whether the DES design will have a role in such

debate, as reduction of strut thickness has been associated with the concern of inadequate radial

force that may likely influence stent expansion and apposition.  

We thank the Reviewer for the interesting comment. We are aware that this represents an important point but we have acknowledged the paucity of data in this setting in the manuscript: “However, specific data on this issue are scant” (Page 15, Lines 7-8). 

  1. In the section “Limitation of Ultrathin DES” the authors state that “stent expansion capacity is more

limited with ultrathin DES than other new-generation DES”. Please provide some reference or

hypothesis to support such statement.

We thank the Reviewer for raising this point. In order to address the Reviewer’s comments, we added a new reference (DOI: 10.1186/s40001-021-00595-7).

  1. There are minor spelling mistakes, please consider careful proof reading before re-submission ) i.e. page 15 “ such has CTO or calcific lesions” should be “as”; in the abstract “ has struts thinner than less than” should be has struts thinner than.

Thank you for this comment. We have provided the requested correction.

  1. Please control list of abbreviation. In the section “Coroflex IsarNeo” the authors used for the first

time the term TLR without providing the extended version.

Thank you for this comment. We controlled list of abbreviation and reported the extended version.

Reviewer #1

The authors reported a state of art review about the current evidence and devices features of the

available ultrathin struts Drug-Eluting Stents. Device characteristics and clinical available data were

reported for the Orsiro, the CoroFlex Isar Neo, the Biomime, the Biomime Morph, the Evermine, the

MiStent and the Supraflex platforms. The topic is of interest and the review is comprehensive. The

paper is nicely written and the manuscript has an interesting structure, with headings referring to the

evidence for each investigated devices and sub-headings referring to specific clinical scenarios that

may be of interest to a large number of readers.

We thank the Reviewer for the careful and thoughtful review of our manuscript.

  1. As per editorial request, and to further improve readers’ interest I suggest adding a paragraph to

explore “Future directions”. In particular, I suggest that Authors should acknowledge the paucity of

data regarding the performance of such devices in real-word scenarios and complex settings as left

main stem disease, in-stent restenosis and coronary bifurcation lesions. The outcomes of the

previous generation DES (namely very-thin strut DES, that is DES with strut thickness < 100 um) was

investigated in some of these contexts along with the impact of specific PCI techniques and of

ancillary techniques such as intracoronary imaging, with interesting results (i.e. see

DOI: 10.1016/j.amjcard.2019.02.013 and DOI: 10.1161/CIRCINTERVENTIONS.119.008325).  

We thank for this comment and akcnowledge the criticism raised by the reviewer.  In this revised version of the manuscript, we added a new section named “Future directions”.

  1. I deem the paragraph “Benefit of strut thickness reduction, a class effect?” nice and of interest.

However, as this is a review focusing on DES features, Authors should more appropriately investigate

current evidence (if any) about structural features differentiating the investigated devices such as

numbers of crowns and connectors, along with their potential impact on outcomes. The impact of

such features was indeed investigated for the previous generation of DES, with fascinating results

(see DOI: 10.1002/ccd.28667).

We thank the Reviewer for this comment. We carefully described structural features of the investigated device in the dedicated section. However, in the updated version of the manuscript, we tried to address this issue in the “ Future directions” paragraph.

  1. The modern PCI era is characterized by an interesting debate about the offsetting relevance of PCI

ancillary techniques, namely invasive coronary physiology vs intracoronary imaging, mainly in

optimizing procedural results and therefore patients’ outcomes. This particularly concern peculiar

setting as ACS ( please see : European Heart Journal, Volume 42, Issue 45, 1 December 2021,

Pages 4656–4668; Catheter Cardiovasc Interv. 2020 Jun 1;95(7):1259-1266. doi:

10.1002/ccd.28410; Int J Cardiol. 2017 Oct 1;244:54-58. doi: 10.1016/j.ijcard.2017.05.108). As

experts in this field Authors should take a position on whether the DES design will have a role in such

debate, as reduction of strut thickness has been associated with the concern of inadequate radial

force that may likely influence stent expansion and apposition.  

We thank the Reviewer for the interesting comment. We are aware that this represents an important point but we have acknowledged the paucity of data in this setting in the manuscript: “However, specific data on this issue are scant” (Page 15, Lines 7-8). 

  1. In the section “Limitation of Ultrathin DES” the authors state that “stent expansion capacity is more

limited with ultrathin DES than other new-generation DES”. Please provide some reference or

hypothesis to support such statement.

We thank the Reviewer for raising this point. In order to address the Reviewer’s comments, we added a new reference (DOI: 10.1186/s40001-021-00595-7).

  1. There are minor spelling mistakes, please consider careful proof reading before re-submission ) i.e. page 15 “ such has CTO or calcific lesions” should be “as”; in the abstract “ has struts thinner than less than” should be has struts thinner than.

Thank you for this comment. We have provided the requested correction.

  1. Please control list of abbreviation. In the section “Coroflex IsarNeo” the authors used for the first

time the term TLR without providing the extended version.

Thank you for this comment. We controlled list of abbreviation and reported the extended version.

Reviewer 2 Report

The authors have done a great job reviewing the different types of drug-eluting stents used in the PCI. However, they can compare the efficacy of each stent. If they can do some sort of meta-analysis, it would greatly enhance the significance of the current review article.

They have discussed the usage of DES in trials involving diabetes mellitus patients. They could also discuss the efficacy of DES in other clinical risk factors.

They have given a generalized limitation for DES; along with that, they could also discuss the limitations of individual DES. 

Author Response

  1. The authors have done a great job reviewing the different types of drug-eluting stents used in the PCI. However, they can compare the efficacy of each stent. If they can do some sort of meta-analysis, it would greatly enhance the significance of the current review article.

We thank the Reviewer for the comment. Currently there are two published metanalysis (Bangalore et al. Circulation 2018, 138 (20), 2216–2226. https://doi.org/10.1161/CIRCULATIONAHA.118.034456 and  Madhavan et al. Long-Term Follow-up after Ultrathin vs. Conventional 2nd-Generation Drug-Eluting Stents: A Systematic Review and Meta-Analysis of Randomized Controlled Trials. Eur Heart J 2021, 42 (27), 2643–2654. https://doi.org/10.1093/eurheartj/ehab280) that were mentioned in the first version of the manuscript.  Both, as we acknowledged in the paragraph “Benefit of strut thickness reduction, a class effect? “,  did not find differences in outcomes according to the type of ultrathin DES used (page 12, lines 15-16 and  lines 22-23).  Therefore performing an additional metanalysis was beyond the purpose of the present manuscript, that aimed instead to summarize existing evidence, focusing on peculiar clinical contexts.

  1. They have discussed the usage of DES in trials involving diabetes mellitus patients. They could also discuss the efficacy of DES in other clinical risk factors.

We thank the Reviewer for this constructive comment. We are aware that this represents a limitation but data on the performance of ultrathin DES in patients with risk factors other than diabetes mellitus are scarce. We have reported in the revised version of the manuscript the available evidence for the Coroflex Isar Neo according to age and sex: “These findings were confirmed at 5 and 10 years follow-up with low incidence of probable/definite ST in both groups and consistent results through pre-specified subgroups of age, gender, diabetes mellitus, and vessel size” (Page 8, Lines 20-21). Of interest, the Fire Trial (Clinical trial.gov: NCT03772743)

will evaluate the performance of the Supraflex/Supraflex cruz in elderly patients (Page 11, Lines 20-24).

  1. They have given a generalized limitation for DES; along with that, they could also discuss the limitations of individual DES. 

We thank the Reviewer for this observation. As discussed above, the available data suggest a class-effect of the ultrathin DES. In the same way their limitations might be class-related, rather than device-related, mainly linked to the reduced strut thickness that could cause an impairment in radial force and expansion capacity. However, we have reported in the manuscript: specific data on this issue are scant. (Page 15, Line 10).